# Wearable Physiological Monitoring System Based on Electrocardiography and Electromyography for Upper Limb Rehabilitation Training

**DOI:** 10.3390/s20174861

**Published:** 2020-08-28

**Authors:** Shumi Zhao, Jianxun Liu, Zidan Gong, Yisong Lei, Xia OuYang, Chi Chiu Chan, Shuangchen Ruan

**Affiliations:** 1Institute of Textiles and Clothing, The Hong Kong Polytechnic University, Hong Kong 999077, China; shumizhao@polyu.edu.hk; 2Sino-German College of Intelligent Manufacturing, Shenzhen Technology University, Shenzhen 518118, China; liujianxun2019@email.szu.edu.cn (J.L.); 15903283r@connect.polyu.hk (Y.L.); chenzhichao@sztu.edu.cn (C.C.C.); ruanshuangchen@sztu.edu.cn (S.R.); 3Department of Electrical Engineering, The Hong Kong Polytechnic University, Hong Kong 518118, China; xouyang@umn.edu; 4Department of Mechanical Engineering, University of Minnesota, Minneapolis, MN 55455, USA

**Keywords:** wearable physiological system, ECG/EMG sensing, upper limb, rehabilitation training, smart wearable device

## Abstract

Secondary injuries are common during upper limb rehabilitation training because of uncontrollable physical force and overexciting activities, and long-time training may cause fatigue and reduce the training effect. This study proposes a wearable monitoring device for upper limb rehabilitation by integrating electrocardiogram and electromyogram (ECG/EMG) sensors and using data acquisition boards to obtain accurate signals during robotic glove assisting training. The collected ECG/EMG signals were filtered, amplified, digitized, and then transmitted to a remote receiver (smart phone or laptop) via a low-energy Bluetooth module. A software platform was developed for data analysis to visualize ECG/EMG information, and integrated into the robotic glove control module. In the training progress, various hand activities (i.e., hand closing, forearm pronation, finger flexion, and wrist extension) were monitored by the EMG sensor, and the changes in the physiological status of people (from excited to fatigue) were monitored by the ECG sensor. The functionality and feasibility of the developed physiological monitoring system was demonstrated by the assisting robotic glove with an adaptive strategy for upper limb rehabilitation training improvement. The feasible results provided a novel technique to monitor individual ECG and EMG information holistically and practically, and a technical reference to improve upper limb rehabilitation according to specific treatment conditions and the users’ demands. On the basis of this wearable monitoring system prototype for upper limb rehabilitation, many ECG-/EMG-based mobile healthcare applications could be built avoiding some complicated implementation issues such as sensors management and feature extraction.

## 1. Introduction

Wearable devices usually involve smart sensors to detect various parameters of the human body and remind the wearer or caregiver to take appropriate action [1,2]. With the advances in mobile technology and the great demand of the aging population for healthcare management, the emergence of wearable medical devices enables people to monitor their personal health information in real time [3,4,5]. Preventing diseases and avoiding emergency health risks are possible due to the feature of continuous monitoring. To date, many wearable healthcare devices provide body biosignals, such as blood pressure, blood glucose levels, body temperature, electroencephalograms, electrocardiograms (ECGs), and electromyograms (EMGs), for diagnosis [6,7,8]. ECG and EMG, which are caused by electrical signal variations during muscular activities, are important and commonly adopted parameters for healthcare management.

During electrocardiography, electrodes are placed on the skin to record the electrical activities of the heart muscles when beating over a period of time [9]. The slight electrical variation on the skin is produced by the electrophysiologic pattern (i.e., depolarizing and repolarizing) of the heart muscle during each heartbeat and detected using the ECG signal detection system. With the increased awareness of people’s health and the continuous development of science and technology, the ECG signal detection system is developed in the direction of miniaturization, family, and intelligence. Sun et al. have integrated conductive fabric ECG electrodes into a health shirt for accurate ECG monitoring during physical exercise [10]. Li et al. have developed a single-arm ECG-based low-power wearable device for heart rate detection during exercise [11]. The wearable ECG monitoring device can provide ECG signals effectively during different types of exercises, activities, and trainings, thereby providing convenience for the rapid and accurate dynamic medical diagnosis.

Electromyography is another electrodiagnostic medicine technique to evaluate and record the muscular electrical signal generated by skeletal muscle activities [12]. Muscle potentials can be produced when muscle cells are activated by electricity or nerves. As such, the biomechanics of human, medical abnormalities, or activation level can be detected and analyzed [13]. Recently, medical and healthcare applications based on EMG signal analysis have emerged [14,15]. A versatile embedded platform for EMG acquisition was proposed by Benatti et al. for gesture recognition [15]. Pradhan et al. analyzed the controlled suitability of assistive devices using a dual-channel EMG biopotential amplifier, and the EMG signals were processed and classified under the artificial neural network [16]. The feature analysis of EMG signal could offer body muscle activity information, such as fitness, fatigue, endurance level, and gesture [17,18].

Poststroke survivors usually undergo long-term physical therapies for rehabilitation because the stroke may result in nerve or muscle disabilities that highly affect their daily activities [19]. Although stroke rehabilitation centers offer physiological activity therapy for stroke survivors under the guidance of therapists, limited resources cannot cover the requirements of each individual [20,21]. Monitored physiological signals were usually explained by trained personnel to improve rehabilitation protocols, but the limited number of therapists and expensive consultation fees result in the delayed rehabilitation process of many stroke survivors [22]. Rehabilitation using hand/wrist robots is effective for the improvement of upper limb functions and could be adopted and operated by patients themselves as home treatment [23,24]. The effectiveness of most existing robot-assisted rehabilitation devices [25] has been assessed via traditional pre- and postclinical evaluations without continuous detection [26,27]. Thereby, a simple monitoring system with a real-time and detailed description of the training progress should be developed for the easy understanding of the treatment-related exercise recovery process for the patients.

In this study, a wearable EMG/ECG sensor-based monitoring system for upper limb rehabilitation training is proposed. The developed system consists of three major components: (1) a wearable monitoring device prototype with sensors (ECG/EMG sensors) and (2) soft robotic glove device, and (3) a software platform (smartphone or laptop with Bluetooth low energy (BLE)) built for upper limb rehabilitation monitoring applications. The ECG/EMG sensors are engineered by leveraging the structure of installation, and the electrodes could tightly contact the skin of users. Small printed circuit boards (PCBs) are used for ECG/EMG signal collection to enhance the wearability. The software uses different platforms (APP and Windows software) to record, display, and analyze the ECG/EMG information in real time, allowing users or caregivers to access the ECG/EMG information easily in terms of graphs and high-level features. Results obtained from the monitoring system could provide rich information for the robotic glove control system to improve the upper limb rehabilitation training protocols. The functionality and feasibility of the developed physiological monitoring system has been demonstrated by adopting a robotic glove system to facilitate rehabilitation training.

## 2. Materials and Methods

### 2.1. ECG System Design and Fabrication

For wearable monitoring applications, a small number of electrodes is desirable for a good fixed installation. A three-electrode configuration, i.e., two active electrodes as the differential inputs of the amplifier and one ground electrode to reduce the harm of current leakage, is typical for ECG monitoring [10]. In this study, two electrodes were adopted to monitor the ECG signal because the two-electrode technique can achieve an isolated circuit to ensure the safety of patients without connecting ground [28]. Two snap fasteners were used to connect the electrodes and the PCB (Figure 1a), and the electrodes adhered to the skin tightly. The PCB had a size of 20 × 65 mm, which was convenient for wearing in practical use. An appropriate monitoring location and good contact between the electrodes and the skin are essential to obtain an accurate ECG signal and reduce the external noise or interference, in other words, stable electrodes ensure good signal effect [29]. Additionally, previous studies have analyzed the effect of human respiration, movement, and the pressure applied by clothing on ECG signals, and results showed that the side area of the chest and the lower right rib area (10th rib bone) are good places with low clothing pressure for stable and high-quality ECG signal detection [10,30].

The PCB integrates a microcontroller, a signal processor, an analog-to-digital (A/D) converter, BLE, and power management modules (Figure 1b). The ECG signals collected by electrodes were delivered to the hardware filter for processing and amplified through the operational amplifier, converted into digital value by using the A/D converter (BMD 101, NeuroSky Company, Wuxi, China), read using the STM32L152 chip (STM32L152, STMicroelectronics, Geneva, Switzerland) in accordance with the communication protocol, and finally transmitted to the smartphone or laptop by using the BLE module.

For the usability requirements of the wearable monitoring system (except the size of the circuit chip), power consumption is another key parameter that is closely associated with the performance of microcontrollers and radio chips [31]. The STM32L152 is adopted as the main chip of the circuit. The STM32L152 has low power consumption, small size, rich peripheral interfaces, and can directly communicate with the BLE module via a serial port [32,33]. The BMD101 chip is designed with an advanced analog front-end circuitry and a powerful digital signal processing structure, thereby targeting biosignal inputs ranging from the μV level to the mV level [34,35]. A bandpass filter with a passing band of 0.5–40 Hz is formed using high- and low-pass filters [36].

### 2.2. EMG System Design and Fabrication

EMG is an electric signal produced by muscle activation and collected from the skin surface [37]. On the lower arm of the human body, approximately 20 muscles work together to achieve the wrist or finger motions [38], and hand rehabilitation exercises include wrist (e.g., supination or pronation) and finger (e.g., fist) movements [16,39]. Two muscles, namely, the flexor carpi radialis (FCR) and the extensor carpi radialis longus (ECRL), were selected for hand motion measurement via palpation. According to previous studies [23,24,40], these muscles are activated weakly in stroke patients. Figure 2 shows that the EMG signals were collected by placing the electrodes on the skin surface of the FCR and the ECRL muscles.

Commercially available disposable electrodes were adopted using the EMG sensors to collect myoelectric signal. Such electrodes have low half-cell potential and can decrease motion artifacts efficiently, which may cause errors in the EMG signal monitoring and recording [41,42]. According to the muscle selection (i.e., the FCR and the ECRL muscles), two active electrodes were placed on the corresponding skin surface of the forearm, and a reference electrode was placed on the elbow joint. These electrodes were connected to the EMG data acquisition (DAQ) board by using shielded wires to avoid the noise caused by radio frequency and electromagnetic interference.

The hardware platform of the EMG system was built by using the STM32L152 chip and precision instrumentation amplifiers (AD8221, Analog Devices, Norwood, MA, USA), and a BLE module [43]. Two groups of EMG signals would be measured from differential electrodes on the FCR and the ECRL muscles. A two-channel block diagram of the EMG hardware test platform for forearm muscle monitoring is presented in Figure 3. The amplitude of the original EMG signals may vary from μV to mV, which is related to the dimension and the depth of the muscles contracting underneath the electrodes [44,45]. The different action potential signals from the FCR and the ECRL muscles are fed to the differential input of the precision instrumentation amplifiers. For the instrumentation amplifier, based on the classic three operational amplifier topology, an error current at the input stage is produced using two constant current biased transistors [46] and fed to precision current feedback amplifiers, and the amplified differential signal and signal from reference electrode can be received by the third operational amplifier at the inputs. Full-wave precision rectifiers, which are integrated for high-frequency noise filtering, are used to rectify the measured EMG signals. Operational amplifiers are applied to amplify the measured EMG signals again [47,48]. EMG signals are delivered to the A/D converter of the STM32L152 chip for digital DAQ, and the obtained data are sent to a remote receiver (such as smartphone or laptop) for the recognition of upper limb activities.

### 2.3. Physiological Monitoring System for Rehabilitation Training

When the EMG and the ECG electrodes are attached to the selected muscles on the upper limb and the chest areas, respectively, the ECG/EMG devices works. The wearable monitoring system is then in operation, and the internal BLE modules are initiated on the circuit boards. The initial setup for calibration parameters was loaded to the registers of the STM32L152 chip. Figure 4 illustrates the software processing framework for the ECG/EMG data. The developed software simultaneously receives ECG/EMG data by connecting with two devices’ BLE modules via two incorporated configuration modes. The ECG or EMG values are read based on a communication protocol. The raw ECG or EMG values are processed and analyzed in accordance with the corresponding mathematical methods (introduced in Section 3), and the time-varying curves are displayed on the screen in real time. Considering the convenience and wide acceptance of smart phones, not only a computer software but also an APP software for the physiological monitoring system were developed to display rehabilitation information to users.

Moreover, the developed physiological monitoring system would be applied in a soft pneumatic robotic rehabilitation glove (SY-HR01C, Shanghai Siyi Intelligent Technology Co., Ltd., Shanghai, China) for hand rehabilitation training demonstration as shown in Figure 5. Monotonous and repetitive palm opening and closing movements [49,50] could be achieved by the robotic glove via pressure variation controlled by a controller (Figure 5b), which consists of a pneumatic pump (KPV04, max pressure 75 kPa, max air mass flow rate 2.0 LPM), a data acquisition control board (STM32 chip) with BLE module, differential pressure sensors (MPX5100DP, Freescale Semiconductor Co., Austin, TX, USA), and valves (SMC V124A, Sintered Metal Corporation, Shenzhen, China). The controller not only uses the fixed mode which set the working time and air pressure through the knob of box, but also connect with a professional software via the internal BLE module to realize the parameter setting. The BLE module receives the control strategy orders from the physiological monitoring system and delivers it to the control board for further analysis (Figure 5c). According to the control strategy, the pump starts producing pressure for the robotic glove to assist hand rehabilitation training. Differential pressure sensors monitor the pressure and offer pressure feedback to achieve precise control. During the rehabilitation training, the upper limb muscles (FCR and ECRL) are monitored by the EMG sensors, and the heart health status are monitored by the ECG sensor, which could real time adjust and give scientific guidance on control strategy for hand rehabilitation training. In this study, a healthy subject with normal hand postures was selected to ensure the reliability and the feasibility of the muscular activity experiment [51,52].

## 3. Results and Discussion

### 3.1. ECG Signal Processing

In this study, the developed software for ECG data processing could perform real-time signal filtering, smooth, feature extraction, display, and rehabilitation training level judgement. First, the high-frequency interference obtained from the electrodes in the ECG monitoring progress was filtered using a 40 Hz low-pass filter. The window median filtering algorithm was adopted without degrading the original ECG signals to reduce the baseline drift [10]. Moreover, the ECG signal obtained using this developed monitoring system may be influenced by the noise caused by body movements. Muscle contractions and unstable contact between the skin and the electrodes during movements directly interfere with the signals and have generated unpredictable artifacts [53]. Therefore, artifacts must be eliminated using the adaptive threshold filter method during ECG signal processing [54,55]. Through the above processing steps, the obtained data nearly approached the actual ECG signal and reliably presented the features (e.g., QRS complex) of the cardiac health situation of patients.

The QRS complex is closely associated with the depolarization of the right and the left ventricles of the human heart and the contraction of the large ventricular muscles, which is the basis of ECG feature analysis. The duration, amplitude, and morphology of the QRS complex are useful in diagnosing disease states (e.g., predicting cardiac arrhythmia and conduction abnormalities) [56,57]. For the QRS complex detection, the baseline data, peak value, and lowest value were taken as the “Q”, “R”, and “S” data, respectively. The wave peak points R, P, T, and the possible clutter peaks were marked to obtain the first-order difference for the ECG data sequence and then find the maximum points. These R, P, and T marks were shown in a section analysis (Figure 6b), which extracted from the raw ECG data plot (Figure 6a). Half of the maximum amplitude value of the detected maximum point was used as the threshold value because the amplitude of the peak value of the R wave was large, and the maximum value other than the peak value of the R wave was excluded. The threshold was set again on the time difference between the adjacent R waves on the basis of the traditional differential threshold method [54], and the singularity was removed by the minimum value of the R–R interval.

The peak points can be detected by analyzing the slopes of the ECG curve, and the heart rate variability (HRV) can be obtained by recording the time difference between adjacent peaks. HRV is used to analyze the heartbeat intervals (i.e., the R–R intervals), providing a passive approach to quantify fatigue [58,59]. The HRV is commonly applied to evaluate the mental workload and the stress of people. For instance, the variation in HRV indicates a decrease in mental workload, which usually occurs among drivers who drive for long periods of time [60,61]. Therefore, the HRV can be extracted in real time via the ECG data processing for fatigue detection during rehabilitation trainings.

A support vector machine (SVM) is a supervised machine learning model that uses classification algorithms to deal with two-group (e.g., excited and fatigue) classification problems, which are trained using various R–R interval signals [62,63]. The initial time interval in experiment is used as a learning phase of SVM for each individual. Generally, the transition from the normal level to the fatigue status was related to a steady decline in pulse rate and associated with an increase in the HRV. When steady values in HRV are observed, the SVM can determine whether the person is experiencing fatigue, and the software system takes the necessary actions, such as alerting. Thus, long-time, safe, and moderate rehabilitation training can be guaranteed on the basis of such applications. In this experiment, one subject (male, 34 year, 76 kg, 170 cm, healthy) was involved as a demonstration example for repeated measurements and analysis. According to his subjective feeling, the excited status usually occurred in the morning, and the fatigue status occurred in the evening, thereby the data of excited and fatigue status were planned to be collected at 10:00 in the morning and 11:00 in the evening, respectively. The ECG data obtained in the excited and the fatigue status could be used to validate the SVM judgement. Figure 7a,b demonstrate the HRV values for the excited and the fatigue status, respectively, and revealed that the pitch of the HRV changes was irregular when the subject was in the excited status and became regular when the subject was in the drowsy status.

The ECG monitoring system can be used to detect long-term human heart physiological characteristics for rehabilitation training through the ECG data processing, feature extraction and analysis, and fatigue algorithm test.

### 3.2. EMG Signal Processing

The obtained EMG signal is a combination of the EMG potentials and the real-time noise and offset, which need to be filtered and processed. Given that the raw EMG data are not suitable for direct analysis, the features of the raw EMG data must be extracted, and other noise should be removed. Among a variety of feature extraction methods (e.g., root mean square (RMS), mean absolute value slope, mean absolute value, slope sign changes, zero crossings, and waveform length) that have been used on different occasions [64,65], RMS is the most effective for real-time human muscular activity estimation. The RMS of the EMG signal can be calculated using the equation:(1)$$\mathrm{RMS}=\sqrt{\frac{1}{N}\sum_{i=1}^{N}v_{i}^{2}}$$
where *N* indicates the segment number (*N* = 200), and *v_i_* is the voltage at the *i*-th sampling. The RMS data were extracted from the raw EMG signal (Figure 8) to validate the algorithm. The feature of the raw EMG signal for the activity level of the muscle to be characterized can be determined using the calculated RMS values. The RMS value could reflect the level of the physiological activities in the motor unit during contraction. For weak muscle activity, the relationship between the activity intensity and the EMG signal had the same trend. For strong muscle activity, the same relationship did not correspond to the activity intensity variations, which needed another calculation method because the signal amplitude of muscle activity was saturated.

However, the signal collected by the EMG module from the skin surface has no baseline, which may bring analysis difficulties due to individual differences. Hence, a normalization process for the EMG signal should be established. The method can obtain the maximum RMS value (RMS_M_) of the EMG data before the rehabilitation training and use the normalization process to calculate the nonnormalized RMS values [48,51]. The equation is presented as follows:(2)$$RMS_{N}=\frac{RMS}{RMS_{M}}$$
where RMS_N_, RMS, and RMS_M_ are the normalized RMS, nonnormalized RMS, and benchmark RMS of EMG (i.e., maximum RMS of EMG value), respectively. A manual testing approach was used to help the tested hand move to the extreme position and obtain the maximum RMS of the EMG value. The test was repeated thrice to reduce the measurement error in the experiment, and the average value was adopted as the RMS_M_.

The activities of certain muscles on different positions produce different EMG signals [14,15]. Therefore, the real-time EMG signals can predict the motions of the upper limb, and the RMS of the EMG signal indicates the level of muscle activities in training. Figure 8 shows the segmentation process of EMG with two-channel monitoring. A group of hand motions (i.e., hand closing, forearm pronation, finger flexion, and wrist extension) were designed (Figure 9a–d), and each motion monitoring was recorded. The RMS of the EMG data and the raw signals collected from the selected FCR and ECRL muscles of the subject were simultaneously displayed (Figure 9e–f). For the action of forearm pronation and finger flexion, the activity intensity of the FCR muscle was greater than that of the ECRL muscle. For the action of hand closing and wrist extension, the activity intensity of the ECRL muscle was greater than that of the FCR muscle. Therefore, according to the EMG characteristics of different muscles and motions, scientific and personalized rehabilitation training programs can be designed for users in accordance with their activity intensity demands.

The EMG DAQ device and data processing method were developed, optimized, and validated using corresponding experiments to provide a discriminatory signal to the user during the upper limb rehabilitation training. This work successfully demonstrated that upper limb muscle activity intensity could be detected and used for EMG monitoring in rehabilitation training.

### 3.3. Physiological Monitoring System Interface

A software platform was designed and developed for the physiological monitoring system by using a multithread technology [66], allowing users to observe the physiological characteristic variation dynamically and in real time toward the upper limb muscle active intensity and the heart activities during rehabilitation training. The mobile application and the computer software were developed for this monitoring system to provide convenience and gain wide acceptance from users.

The developed mobile application interface (Figure 10a) consisted of two graph panels and five main functional tabs, including “Scan”, “Send”, “Save”, “Clear”, and “D-wave”. The Scan function, the first tab, is for scanning the broadcasting BLE devices and establishing connection with one. The Send function, the second tab, is for communicating with the respective PCB for the user’s control parameters transmission (such as data collection rate, glove working cycles, etc.). Data can be saved using the Save function and cleaned using the Clear function. The D-wave function, the fifth tab, is for finding the peaks of the data curves. Graph panels include the data display and the graph depict function. The graph depict plots a view of the accumulative ECG or EMG data along with the time to visualize the relevant physiological characteristics variation of subject in real time. In this program, two timers were responsible for working smoothly. One timer was responsible for receiving data, and the other timer was responsible for processing and display.

For the computer software, a user-friendly interface with a customized graphical user interface was designed. Various function buttons and control modes are available for users to set customized rehabilitation monitoring parameters. The interface involves four different areas (Figure 10b) (1) the parameters setting area, where users can input the data collection rate to control data delivering frequency; (2) the button-controlled area for special action control; (3) the data display area to present real-time monitored information in curves; and (4) the calculation display area for showing the key monitoring parameters (such as heart rate, muscle intensity average values). A two-graph panel was designed to display the real-time graphical illustrations of the EMG and the ECG data.

The software system is described with the function of monitoring, visualizing, and recording detailed ECG/EMG data. Soft robotic glove control parameters could also be input on the software interface. Furthermore, more physiological parameters such as physical activity, and threats to safety, etc. could be revealed by the collected data and offered reminding [2,3,11]. Thus, a physiological monitoring system composed of the software platform and hardware device for rehabilitation training has been established successfully.

### 3.4. Upper Limb Rehabilitation Application

To further evaluate the feasibility and usability of the developed physiological monitoring system in practical use with the implemented device, a soft robotic rehabilitation glove was adopted as an application for testing. The experiment was carried out in two situations—when the subject (good sleep last night, get up at 7:00 in the morning, and keep awake till evening test) was in the excited status and in the fatigue status. Data of the excited status were collected at 10:00 am, while data of the fatigue status were obtained at 11:00 pm. EMG data were collected from selected muscles at three different stages for analysis as shown in the Table 1.

According to ECG monitoring, a significant variation occurred in HRV when people started with an excited status, and the HRV variation was small when people started with a fatigue status. Although the number of subject and relevant data were limited and cannot represent all, such findings have already been demonstrated by previous studies [59,60,61]. With regard to the EMG analysis, when comparing the variation amplitudes between the excited and fatigue status, the intensities of muscle activity in excited status were greater than that in fatigue; when comparing the variation amplitudes between the first and last training bout, the intensities of muscle activity were perceived to significantly decrease along with the training time. These findings indicated that the muscle performance level was closely associated with the psychological fatigue level that may significantly influence the rehabilitation training effects. Thus, it is better to undergo rehabilitation training in the excited status and adopt the developed system for real time physiological monitoring to ensure the training efficacity. Traditional glove assisting training without physiological monitoring could not provide adaptive control for the glove to improve rehabilitation training, as shown in Figure 11a. To optimize control performance of the robotic glove for improving rehabilitation training and refer to some related adaptive control methods in similar studies [67,68,69,70], an adaptive control strategy was induced into the glove control system as presented in Figure 11b. For instance, if the HRV variation of 10 consecutive measurements less than 5 ms was considered as the fatigue level which matched the subjective judgement of the subject, the ECG sensors could then easily determine whether the user is in the excited or fatigue status based on the monitored data. In this case, once the user was in the fatigue status, the pressure of the pneumatic glove would be weakened to protect muscles from secondary injuries, and when the user was in the excited status, the pressure would be strengthened to increase training intensity gradually. Meanwhile, the intensity of muscle activity could also be predetermined as control parameters (e.g., ECRL muscle activity intensity = 1.5 V) to perform a certain motion task. According to the monitored intensity variation of muscle activities, the difference between the predetermined and detected EMG data could offer adaptive commands to the pneumatic robotic glove for pressure adjusting in order to maintain the predetermined performance level. Thereby, the developed training device with an adaptive control strategy could enable users to perform rehabilitation exercises easily and sustain the training endurance.

The results illustrated in Figure 12 showing that without adopting this adaptive control strategy, the intensity of muscle activity decreases with time, and after using the adaptive control strategy, the intensity of muscle activity could be maintained at a required level to ensure and sustain the rehabilitation training effectiveness. Although there are some similar systems for muscle activity monitoring and rehabilitation [51,71,72], few studies developed a real time wearable system with simple operation, and traditional methods indicate the difficulty to effectively apply intensive training programs without real time monitoring [47]. With the purpose to improve rehabilitation effects, it is important to ensure that the involved patients are in the excited status for repetitive movements in order to stimulate neuroplasticity [73]. Therefore, the developed real time physiological monitoring system was meaningful, which could visualize the training effects and help to assist and adjust training for a long period of time. In the future, more subjects would be recruited in clinical trials to further explore the treatment effects of this system-based robotic glove for upper limb rehabilitation and to offer an optimal rehabilitation training protocol for each individual.

## 4. Conclusions

This study demonstrates a newly developed wearable physiological monitoring system which integrated the ECG/EMG sensors and the DAQ boards to obtain accurate physiological signals during upper limb rehabilitation training. A robotic glove was adopted as an applied example of the developed system to assist upper limb rehabilitation training, and the commonly adopted hand activities were monitored using EMG sensors, while the physiological status of people (from excited to fatigue) was detected by the ECG sensor. The ECG/EMG signals obtained from the electrodes were filtered, amplified, digitized, and then transmitted to a remote receiver (i.e., smart phone or laptop) via the BLE module for further analysis. A software platform integrating several algorithms (i.e., SVM algorithm for excited and fatigue judgement, RMS algorithm for EMG analysis, adaptive algorithm for rehabilitation training control) for data analysis. Moreover, this platform was combined with a robotic glove control module, which could provide an accurate control strategy to the robotic glove for rehabilitation training improvement. The information of ECG/EMG, such as heart rate, heartbeat intervals, and ECRL and FCR muscle intensity, was real time displayed on the interface of software, which provided a large amount of medical reference information for doctors or patients. We demonstrate the highly-integrated and multifunctional physiological monitoring system which can be used to assist the robotic glove improving upper limb rehabilitation training. The feasible result provided a novel technique to monitor individual ECG and EMG information holistically, which can be potentially applied in upper limb rehabilitation training in accordance with the specific treatment condition and the users’ demands. Based on this wearable EMG and ECG system prototype, many ECG/EMG-based mobile healthcare applications can be built to avoid complicated implementation issues, such as sensor management and feature extraction, and may considerably improve the effectiveness of home treatment.

## Figures and Tables

**Figure 1 sensors-20-04861-f001:**
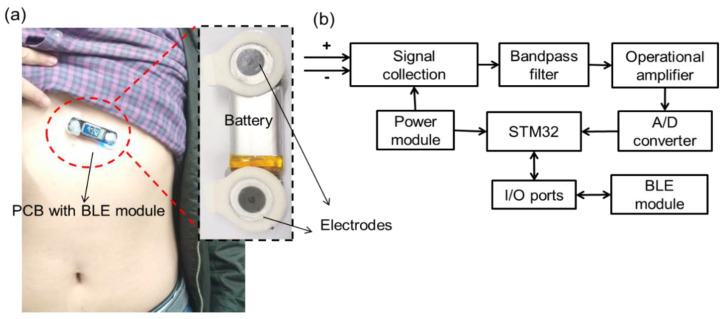
Overall architecture of the wearable electrocardiogram (ECG) monitoring system: (**a**) ECG device placed on the side area of the chest and the lower right rib area (10th rib bone); (**b**) working flowchart of the ECG hardware.

**Figure 2 sensors-20-04861-f002:**
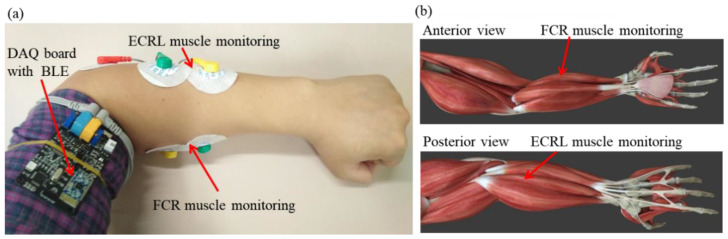
Monitoring of muscular activities: (**a**) monitoring of the extensor carpi radialis longus (ECRL) and the flexor carpi radialis (FCR) muscles; (**b**) muscle distribution.

**Figure 3 sensors-20-04861-f003:**
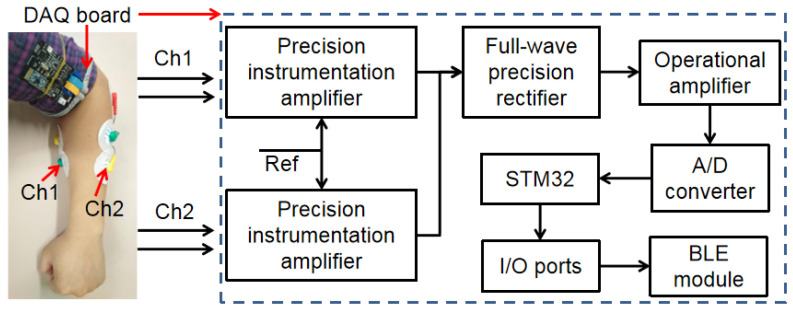
Two-channel block diagram of the electromyogram (EMG) hardware platform.

**Figure 4 sensors-20-04861-f004:**
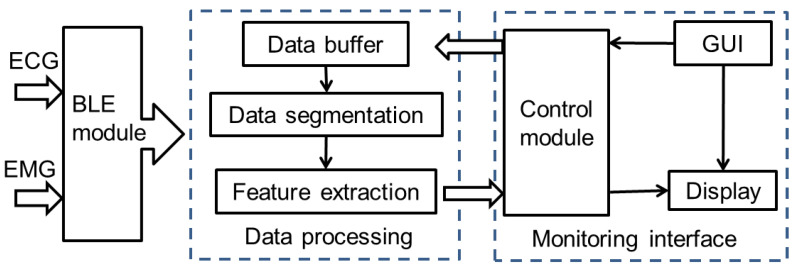
Software processing framework for the ECG/EMG data.

**Figure 5 sensors-20-04861-f005:**
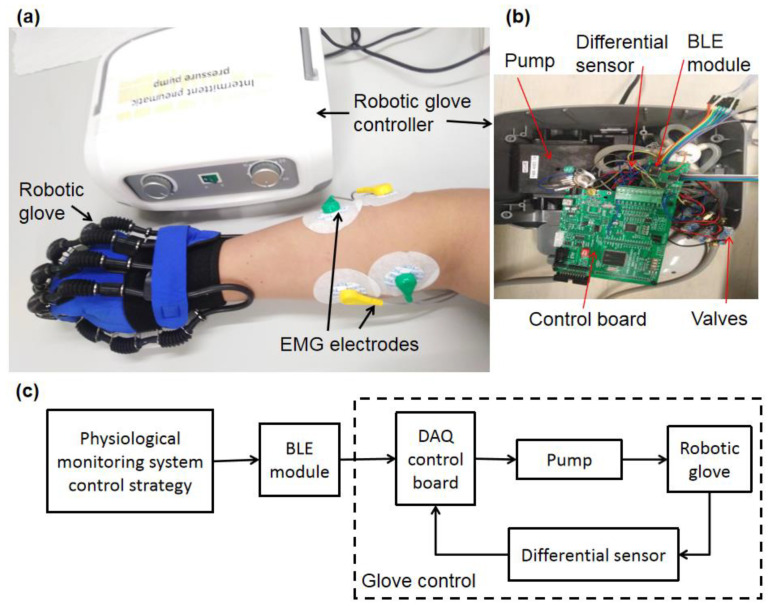
The physiological monitoring system integrating into a pneumatic robotic glove for hand rehabilitation training: (**a**) the robotic glove system; (**b**) the controller; (**c**) the work flowchart.

**Figure 6 sensors-20-04861-f006:**
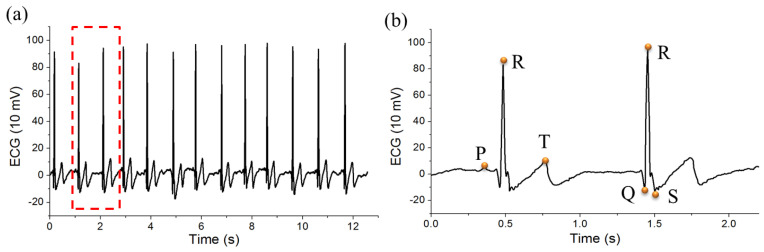
ECG signal analysis: (**a**) raw ECG data plot; (**b**) ECG graph for QRS detection.

**Figure 7 sensors-20-04861-f007:**
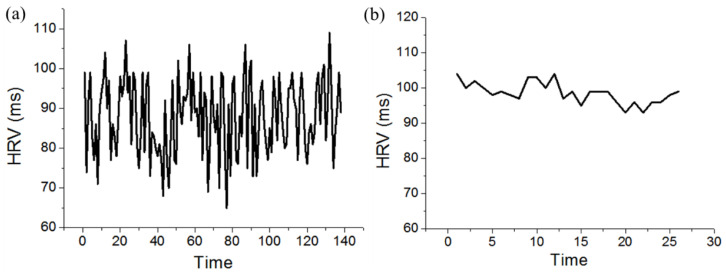
Heart rate variability (HRV) values for activity analysis: (**a**) excited status; (**b**) fatigue status.

**Figure 8 sensors-20-04861-f008:**
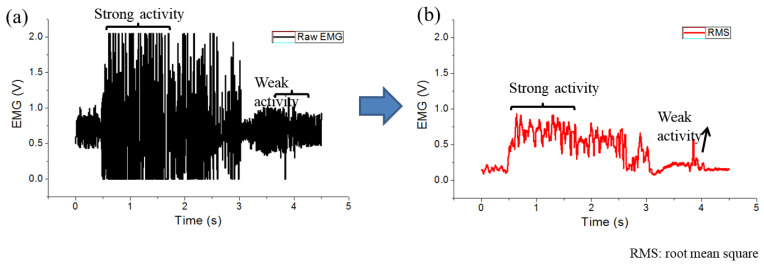
EMG signal processing for RMS analysis: (**a**) raw EMG signal; (**b**) RMS of EMG signal.

**Figure 9 sensors-20-04861-f009:**
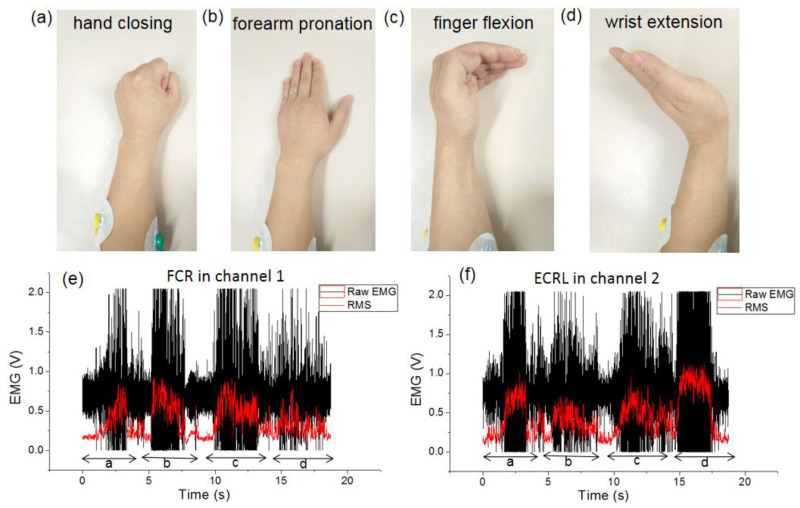
Segmentation process of EMG with two-channel monitoring in the FCR and the ECRL muscle positions: (**a**–**d**) designed hand motions; (**e**) FCR muscle monitoring in channel 1; (**f**) ECRL muscle monitoring in channel 2.

**Figure 10 sensors-20-04861-f010:**
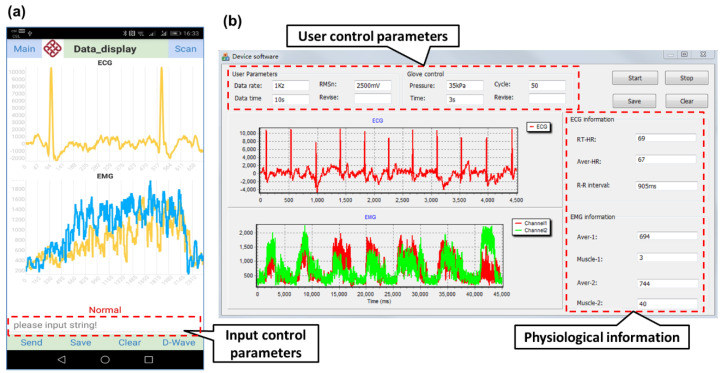
ECG/EMG monitoring software: (**a**) mobile interface for ECG/EMG monitoring (robotic glove control use control parameters); (**b**) computer interface for ECG/EMG monitoring and robotic glove control.

**Figure 11 sensors-20-04861-f011:**
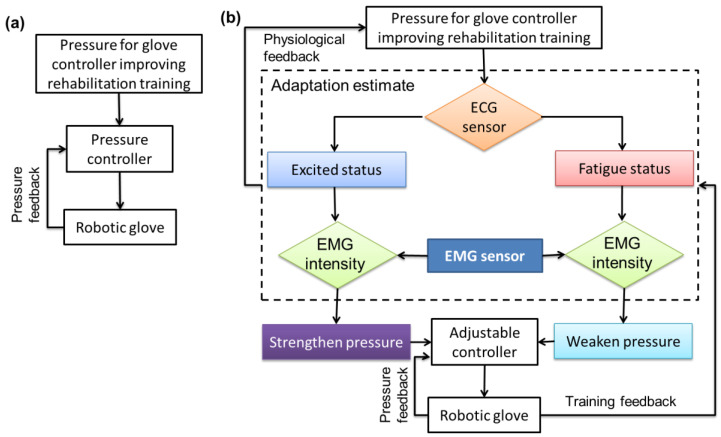
Robotic glove pressure adjusting for rehabilitation training (**a**) without and (**b**) with an adaptive control strategy.

**Figure 12 sensors-20-04861-f012:**
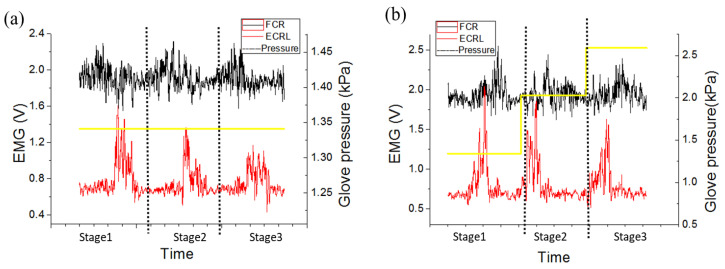
Muscle intensity monitoring for robotic glove controlling (**a**) without and (**b**) with an adaptive strategy.

**Table 1 sensors-20-04861-t001:** Physiological data of a subject in different time intervals.

Time Interval	Excited Status			Fatigue Status		
	ECRL (RMS)	FCR (RMS)	ECG (HRV)	ECRL (RMS)	FCR (RMS)	ECG (HRV)
1st min	2.0 ± 0.4 V	1.6 ± 0.4 V	101 ms	1.8 ± 0.3 V	1.5 ± 0.3 V	96 ms
30th min	1.8 ± 0.3 V	1.5 ± 0.4 V	80 ms	1.5 ± 0.2 V	1.4 ± 0.2 V	99 ms
60th min	1.6 ± 0.4 V	1.5 ± 0.3 V	113 ms	1.3 ± 0.3 V	1.5 ± 0.2 V	98 ms

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
