# Peer review of "Wearable Physiological Monitoring System Based on Electrocardiography and Electromyography for Upper Limb Rehabilitation Training"

_sensors, 2020, doi:10.3390/s20174861_

Round 1
Reviewer 1 Report
This paper has been revised according to my comments.
Author Response
Thanks for taking your precious time reviewing this manuscript.
Reviewer 2 Report
The paper has greatly improved by the previous edit rounds.
Some language editing is still required, in particular the newly added paragraphs suffer from sentences like
"In the future, more subjects could be involved to test and give a quantitative percentage for improvement relation from the medical rehabilitation perspective."
which which are almost incomprehensible.
Reviewer 3 Report
The authors have made a significant effort to improve the paper. However, the paper has some deficiencies that I indicate below:
In line 88 the paper indicates: “Results obtained from the monitoring system could provide feedback to the robotic glove for improving the upper limb rehabilitation training protocols”. This feedback is performed (line 371): “… .To optimize training performance of the adopted pneumatic robotic glove, the training was equipped with the function of adaptivity for rehabilitation. Adaptivity means to adjust the training tasks automatically according to dynamic hand performance in rehabilitation progress ”. But this adaptive strategy is “empirical”, and is based on (page 377): ”…. if the HRV variation of 10 consecutive measurements less than 5ms was considered as the fatigue level, the ECG sensors could then easily determine whether the user is in excited or fatigue status based on the monitored data ”. Why this criterion, and not another, has been chosen for this adaptive strategy? Is there any scientific evidence for choosing this strategy?
I believe that the adaptive strategy that is implemented in a rehabilitation system should be more grounded, and better defined from a technical point of view.
On pages 423-425 it is stated that: “The feasibility and usability of the developed physiological monitoring system for upper limb rehabilitation training was validated by adapting an adaptive method for assisting robotic glove improving upper limb rehabilitation training”. This statement is not correct, a rehabilitation device and a control strategy cannot be validated when it has been applied TO A SINGLE PATIENT (as indicated on page 256: “one subject (male, 34 yr, 76 kg, 170cm, healthy) was involved as a 256 demonstration example for repeated measurements and analysis ”). Furthermore, it is not indicated whether this person-patient has some kind of disability, or is a healthy subject.
Therefore, it is necessary to carry out new tests with a greater number of people or patients, so that, from the analysis of the results, it can be determined whether the control strategy is valid or not.
Therefore, I believe that the paper has a correct first part in which it describes the sensors used and the analysis of the data provided by these sensors, but the part corresponding to the adaptive control strategy and tests with patients must be improved and increased. .
Author Response
Please see the attachment.

This manuscript is a resubmission of an earlier submission. The following is a list of the peer review reports and author responses from that submission.
Round 1
Reviewer 1 Report
This paper developed a wearable physiological monitoring system based on electrocardiography and electromyography for upper limb rehabilitation improvement. However, the novel contribution of this research is unclear. The ECG sensor and EMG sensor are very common and have been presented in a lot of existing researches. Besides, the EMG processing algorithm used for hand posture detection is not innovative. Moreover, the authors are required to explain how to apply the proposed system to improve upper limb rehabilitation in more detail. The effectiveness of the proposed system should be compared with other similar research, including the advantages and disadvantages.
Reviewer 2 Report
The paper describes in sufficient detail the approach take by the authors to develop an ECG/EMG based system for monitoring during rehabilitation. What is missing though, is ANY information about whether the system actually achieves the goals of comprehensive monitoring, providing actionable feedback during exercise and a good usability.
In short, the approach looks promising, but any validation information is missing. The conclusion "The feasible result has provided a novel technique to monitor individual ECG and EMG information holistically" is thus, by and large, unfounded.
This is really deplorable and the reviewer would have liked to see
- either some information backing the claim that the information generated is actually helpful, or
- the article rewritten in a form that makes clear that this is an approach that has been hitherto untested and lacks for the time being any validation.
On a different token, the paper was unecessarily hard to read because the choice of grammatical tenses seemed to be haphazard: Instead of "was" or "were" one reads "is" or "are" or the other way around. This is not a minor point because the reader is left unsure about what has actually been achieved and what is still unfer development.
For instance, the sentence
"The EMG DAQ device and data processing method are developed, optimized, and validated using corresponding experiments to provide discriminatory signal to the user during the upper limb rehabilitation training."
seems to say that this is an ongoing activity, but the reviewer assumes that this already has been done.
Reviewer 3 Report
This paper presents a wearable monitoring device for upper limb rehabilitation that integrates EMG and ECG sensors.
The paper is well structured. It presents in sections 2, 3.1 and 3.2, data collection and processing. These sections are correct, they present data collection and processing methods that are commonly used by researchers working in the field of EMG and ECG. Therefore, these sections DO NOT have any scientific contribution.
Section 3.3 of the paper presents an user interface software that allows viewing and recording the ECG and EMG signals taken from the patient. This section also has no relevant scientific contribution.
The title of paper is: “Wearable Physiological Monitoring System Based on ECG and EMG for Upper Limb Rehabilitation Improvement”. This title seems to predict a relationship between the “developed EMG and ECG signal monitoring system”, and “an improvement of upper limb rehabilitation process”. There is no evidence in the paper to indicate that the development presented can contribute to an improvement in the rehabilitation process.
Lines 281- 283 appears: “Therefore, different rehabilitation activities correspond to different EMG signal characteristics, and scientific and personalized rehabilitation training programs can be designed for users in accordance with these activity characteristics”. I do not understand this statement, and I cannot find its justification in the paper.
The statement on lines 291-292 is not justified in the paper: “This work demonstrates successful upper limb muscle activity intensity monitoring system for EMG rehabilitation training”.
For the same task (for example, closing / opening the hand), level and the shape of the EMG signals is very dependent on the person. Therefore, when working with sEMG, it is necessary, for each person, to establish a baseline level at the beginning of taking sEMG. This is not discussed in the paper. The authors only indicate (lines 275-277) that they have stores a group of hand motions (hand closing, forearm pronation, ...), but they do not indicate what they will use this stored EMG data for each movement.
If this paper tries to “IMPROVE” upper limb rehabilitation using EMG and ECG, the paper must include a device for rehabilitation of upper limb, where you can test how EMG and ECG monitoring and processing can improve therapies. That is, it will be necessary to develop and implement device control strategies with which the upper limb rehabilitation is performed, so that it can be considered that the EMG and ECG signals contribute to the improvement of rehabilitation.
In summary, this paper only presents techniques (already known) for taking EMG and ECG signals, processing techniques (already known) for that data, and graphical representation of the signals taken. It does not provide any research in these fields. It does not develop control strategies for the rehabilitation sessions, which, based on the processing and analysis of EMG and ECG, can contribute to improving upper limb rehabilitation.
Round 2
Reviewer 1 Report
The authors have revised the manuscript according to some of my comments. However, the novel contribution of this paper is still unconvincing.Reviewer 2 Report
I appreciate the authors' efforts to report practical results from the application of the developed approach.
However, from a methodical point of view, the given information cannot be used as substantiating the claims:
- No information is given about the conditions the subjects were in, the rehabilitation regime or even the number of subjects.
- No quantitative information is given in terms of superiority to conventional approaches. In particular, there doesn't seem to have been a control arm.
So basically my initial complaint that such an interesting approach deserves a well-designed study to substantiate the claims made. I would strongly encourage the authors to maybe focus on a single topic like "Adaptivity" and provide quantitative information about the effects thereof (e.g. number of subjects who had to cancel training early or the like).
As for the English: There are slight improvements, but still a general mixture of tenses, making comprehension harder than it should be. On a general note: Descriptions of the system's architecture or operations should be in present tense.
For example
"The software used different platforms (application and Windows software) to record, display, and analyze the ECG/EMG information in real time"
should become
"The software uses different platforms (application and Windows software) to record, display, and analyze the ECG/EMG information in real time.
And
"The ECG signals collected by electrodes were delivered to the hardware filter"
should become
"The ECG signals collected by electrodes are being delivered to the hardware filter"
There are still numerous cases of this pattern in the manuscript. A similar thing could be said about sentences like
"In this study, a wearable EMG/ECG sensor-based monitoring system for upper limb rehabilitation training was proposed."
which should become
"In this study, a wearable EMG/ECG sensor-based monitoring system for upper limb rehabilitation training is being proposed." or, preferably
"This study proposes a wearable EMG/ECG sensor-based monitoring system for upper limb rehabilitation training."
Reviewer 3 Report
First, I thank the authors of the paper for the corrections made. With these corrections the paper has improved.
Once the new version of the paper has been read, I consider that the following should be improved-modified:
- Lines 188-189 indicate: "the physiological monitoring system was 188 integrated into a soft pneumatic robotic rehabilitation glove". If the robotic device is a commercial device, indicate its name. If it is a prototype robot developed in the laboratory, it must be indicated.
- In the new section 3.4, on lines 338-339, it is stated: "The experiment was carried out in two situations, when the subject in excited 338 status and in fatigue status". It should be indicated: how many people participate in the experiment, percentage of men and women, the mean age and average weight.